# Treatment Effect Estimation to Guide Model Optimization in Continual Learning

**Jonas Seng[1] [†],   Florian P. Busch[1,2],   Matej Zečević[1],   Moritz Willig[1]**

[1]Computer Science Department, AIML, TU Darmstadt, Germany,     [2]hessian.AI
[†]correspondence: `jonas.seng@tu-darmstadt.de`

## Abstract

Continual Learning systems are faced with a potentially large numbers of tasks to be learned while the models employed have only limited capacity available, which makes it potentially impossible to learn all required tasks within a single model. In order to detect on when a model might break we propose to use treatment effect estimation techniques to estimate the effect of training a model on a new task w.r.t. some suitable performance measure.

## Motivation

Continually learning new concepts and solving new tasks is one key element of human intelligence which accompanies us throughout our entire lifespan and seems to be more important than ever in these accelerated times of technological progress. For instance, the rising dynamics of the job market demands employees for continually learning, e.g. using a new software being introduced in a company, while not forgetting how to solve problems we face all the time during work, e.g. communicating properly with new customers. Continual Learning (CL) aims to transfer this ability to Machine Learning (ML) to obtain models which are capable of adapting to new tasks without losing the ability to solve tasks seen earlier. Among others, this comes with a set of benefits: (1) existing knowledge gathered by learning a sequence of tasks can be exploited for reaching better performances and making models more robust by leveraging similarities among tasks and (2) continually updating models avoids the need to fully re-train once a new task is faced, thus CL helps making ML more resource-efficient (Delange et al. 2021; Parisi et al. 2019; Mundt et al. 2020). In CL, a task can correspond to one of the widespread problem-definitions used in ML, i.e. supervised learning, unsupervised learning or combinations thereof. One of the most prominent problems in CL is catastrophic forgetting which describes the observation that ML-models (especially Neural Networks) tend to *forget* about tasks they have learned previously once they are trained to solve a different task (McCloskey and Cohen 1989; Delange et al. 2021; Parisi et al. 2019; Mundt et al. 2020). Methods aiming to overcome this issue either train expert-models for each task, replay old data while training on new tasks or fix certain parameters in the models

which are considered to be important to solve tasks that can be solved already. Expert-models suffer from high resource consumption and they are not able to exploit old knowledge due to isolated parameter-sets per expert. Though replay-based approaches also suffer from high memory consumption, they are widely used because such approaches have shown good performance. (Kirkpatrick et al. 2017; Rebuffi, Kolesnikov, and Lampert 2016; Mundt et al. 2020; Delange et al. 2021). Fixing parameters which are deemed important to solve former tasks is less resource-intensive, thus being a reasonable approach as well. However, for all approaches which share parameters across tasks, questions like the following arise: "Does the model have enough capacity to learn a new task?" Given a paramerterized model, which effect will training on a new task have w.r.t. the overall model performance? Given that we have trained our model on a sequence of tasks, what would be the state and performance of our model if we had not trained on the last $k$ tasks? Answering such questions is crucial in order to have guarantees w.r.t. model performance and robustness. Also it increases flexibility of CL-systems since answering such questions allows to determine when model-complexity has to be increased. Estimating effects in counterfactual settings enables CL-systems to find proper trade-offs, e.g. when we *have to* learn a new task, but there is not enough capacity, i.e. we are sure that the overall model-performance will decrease. Then, with counterfactual reasoning, one could identify knowledge in the model which causes the lowest decrease in performance once this knowledge is discarded to make space for the new task to be learned. To see why estimating the effect of training on a new, unseen task has, consider the following example: Assume a robot that already has learned to walk in an environment and to jump over obstacles. Now, it is confronted with learning to collect certain items while moving through the environment. However, it does not have enough capacity to learn all three tasks. Being able to predict that learning to collect items will lead to bad performance in e.g. jumping over obstacles (that might harm the robot itself), allows the robot to decide to not learn the new task and stay safe instead.

A robust and well known framework to compute the effect (here: model performance) that causes (here: training on a task) have in some given system is *treatment effect estimation (TEE)*. Since TEE is theoretically well understood and

Published at AAAI 2023 Bridge on Continual Causality.

widely used (e.g. for assessing the causal effect of new drugs in medicine (Bica et al. 2021)), we want to make use of its robustness and employ it to reason about the effects of continual training on ML models.

## Treatment Effect Estimation

TEE has its grounding in Causal Inference. The goal is to estimate the effect of an intervention in a system on some variable (Becker and Ichino 2002; Imbens 2004; Rubin 2005). Besides the Potential Outcome framework of Rubin (2005), the *do*-calculus proposed by Pearl (2009) is a strong framework which can be used to compute entities required for TEE. The *do*-calculus is able to capture asymmetries rendered by causal structures (i.e. if $A$ is the cause of $B$, changing $A$ changes $B$ but not vice versa). Following this rationale, the average treatment effect (ATE) of a binary variable $X$ on a variable $Y$ can be defined as follows:

$$\text{ATE} = \mathbb{E}[Y|do(X=1)] - \mathbb{E}[Y|do(X=0)] \quad (1)$$

ATE is just one of many treatment effect quantities one can estimate/compute, another important quantity is the individual treatment effect (ITE) where one focuses on the outcome of an individual system configuration instead of taking an expectation (Tabib and Larocque 2019). However, we will focus on ATE here. It is also possible to consider counterfactual scenarios: Instead of asking how the system will behave under an intervention, we ask how the system *would have* behaved if an intervention was performed (Hsu, Lai, and Lieli 2022; Yao et al. 2021).

## Connecting TEE and CL

In order to perform TEE, we have to know which variable is caused by which other variable(s). We assume that a causal graph is known or can be designed by hand. For example, Figure 1 shows a causal graph of one "step" in a CL-system: $t_i$ denotes a task we obtain at step $i$, $\tau_i$ is a binary decision variable indicating whether we update our model based on $t_i$, $\theta_{i-1}$ and $\theta_i$ are the model-parameters at step $i-1$ and $i$ respectively, $l_i$ is the model-performance w.r.t. all tasks at step $i$ and $T_{i-1}$ refers to the set of all tasks we have trained on until step $i$ (excluding task $i$). Note that all variables except for $l_i$ are independent of $T_{i-1}$ since we observe $\theta_{i-1}$ which represents the accumulated knowledge over $T_{i-1}$, thus older tasks are not needed to estimate these variables.

**TEE in Factual Settings** Sticking with the example in Figure 1, a natural question to be answered is: Obtaining a new task $t_i$, will the average model performance $l_i$ significantly decrease when updating the current parameters $\theta_{i-1}$ on $t_i$? Formally this question corresponds to estimating the *conditional average treatment effect (CATE)* $\mathbb{E}[l_i|do(\tau_i = 1), t_i, \theta_{i-1}] - \mathbb{E}[l_i|do(\tau_i = 0), t_i, \theta_{i-1}]$. Estimating this quantity requires us to estimate the case where $do(\tau_i = 1)$ only since $do(\tau_i = 0)$ can be approximated by evaluating the current model on all tasks and average the performance. Estimation of $do(\tau_i = 1)$ case can be done with a 2-step-procedure: First, estimate a distribution over $\theta_i$ s.t. the parameters that would result from training on $t_i$ have

high probability, denoted by $p(\theta_i|t_i, \theta_{i-1})$. Once this distribution is estimated, the expectation of $l_i$ can be computed by:

$$\int_{l_i} \int_{\theta_i} l_i \cdot p(l_i|\theta_i) \cdot p(\theta_i|t_i, \theta_{i-1}) \quad (2)$$

Estimating distributions $p(\theta_i|t_i, \theta_{i-1})$ and $p(l_i|\theta_i)$ comes with the advantage of being able to quantify the uncertainty in the parameter-prediction. However, since especially $p(\theta_i|t_i, \theta_{i-1})$ is likely to be a complex distribution, computing moments of this distribution is probably resource-intensive because approaches like Monte Carlo have to be used. Instead one could perform point estimates of $\theta_i$. For this a promising starting point could be to utilize Influence Functions (Koh and Liang 2017). These measure the effect of a single sample of a dataset on the model-parameters. In our setting Influence Functions would measure how $\theta_i$ would differ if we would have trained on task $t_i'$ instead of $t_i$ where $t_i'$ is a version of $t_i$ with one sample dropped and parameters being initialized to $\theta_{i-1}$ before training. To estimate the effect of an entire task on $\theta_i$, one would have to extend Influence Functions to estimate the effect of an entire dataset on the parameters. Once the effect of $t_i$ on the parameters can be computed, evaluating the effect of task $t_i$ on some performance metric is straightforward. Knowing the effect of training on a task $t_i$ w.r.t. some performance measure allows to determine when the model should be equipped with additional capacity, e.g. by adding more parameters. Additionally, the estimated change in parameters can be used to warm-start the next training-stage.

**TEE in Counterfactual Settings** Another issue we are confronted with in CL-settings is the following: Assume we have a fixed resource-constraint (i.e. our model has a maximum possible capacity) and we obtain a new task which will decrease the overall model performance. Then we have to identify those parts of knowledge represented by our model which will cause the lowest decrease in performance. This can be considered as identifying the task that contributes the lowest amount of knowledge to our model, which in turn can be formulated as a counterfactual question: *"What would the model performance be if we had not trained on $t_{i-k}$ but on $t_i$?"* This question can be answered by estimating a series of ATEs in counterfactual settings s.t.

$$\mathbb{E}[\overline{l_i}|do(\tau_i = 1), t_i, \overline{\theta_{i-1}}] - \mathbb{E}[l_i|do(\tau_i = 1), t_i, \theta_{i-1}] \quad (3)$$

is maximized where $\overline{l_i}$ and $\overline{\theta_{i-1}}$ corresponds to the value of $l_i$ and $\theta_{i-1}$ respectively if $\tau_{i-k}$ had been 0, i.e. if we had not trained on $t_{i-k}$. Having such a method to estimate $\overline{l_i}$ and $\overline{\theta_{i-1}}$ would not only allow for assessing which knowledge does not contribute much to the overall model-performance, it also can be used to warm-start the model once the knowledge causing the lowest performance-drop if discarded was identified. Again, as in the factual setting, one could aim to estimate a distribution over model parameters and performance-measures. Of course, one faces the same challenges as in the factual setting when estimating complex distributions and their moments. As above, an alternative approach is to use point-estimates in the form of Influence Functions. However, instead of only extending them to

capture the change in model parameters $\theta_{i-1}$ of a task $t_i$, another extension would be necessary to capture the change of a sequence of tasks. For example, say we have continually trained a model on 4 tasks and obtain a 5th task. To estimate the parameters' values if we would not have trained on e.g. task 3, we first would have to "undo" the changes of task 3 and 4. Then $\theta_5$ has to be estimated based on $\hat{\theta}_2$ where $\hat{\theta}_2$ is an estimation of $\theta_2$ based on rolling back changes made by task 3 and 4. Assuming we are allowed to save $\theta_i$ at each step $i$ and have access to each $t_i$ or a representative thereof (as in pseudo-rehearsal), the above problem simplifies to estimating the effect of a sequence of tasks, i.e. it reduces to solving a sequence of the same problem as in the factual setting.

**Robot-Example**   To show that CL can benefit from estimating the effect of training a model on a certain task, we return to the motivational example from above: Assume $t_1$ corresponds to the task *walking*, $t_2$ corresponds to the task *jumping over obstacles* and $t_3$ corresponds to the task *collect items*.
In the factual case, given that the robot learned to solve $t_1$ and $t_2$, we can estimate the overall effect of learning $t_3$:

$$e(t_3) = \mathbb{E}[l_3|do(\tau_3 = 1), t_3, \theta_2] - \mathbb{E}[l_3|do(\tau_3 = 0), t_3, \theta_2]$$

Since $l_3$ denotes the performance across all tasks, a simple decision rule could be to train on $t_3$ if $e(t_3) > 0$, i.e. the overall gain when training on $t_3$ is higher than possible performance drops in single tasks. Since walking and jumping are substantially different than collecting items, it is likely that $e(t_3) > 0$, thus the robot would decide to learn $t_3$ if it has enough capacity for $t_3$.
For counterfactual cases assume that the robot had the capacity to also learn $t_3$ from above and we obtain yet another task $t_4$ in which the robot should learn to search for an energy supply-station once its battery is low. Assume the robot's model driving its decisions has not enough capacity to learn $t_4$. However, it is forced to learn $t_4$ since without solving it the robot will stop working once its battery is empty. Further assume a task $t_i$ does not carry information about a different task $t_k$, i.e. learning $t_i$ does not help us solving $t_k$. Now the robot has to identify the task creating the least harm to its overall performance which can be done by estimating:

$$\mathbb{E}[\overline{l_i}|do(\tau_i = 1), t_i, \overline{\theta_{i-1}}] - \mathbb{E}[l_i|do(\tau_i = 1), t_i, \theta_{i-1}]$$

Here, $\overline{\theta_{i-1}}$ denotes an estimation of the model-parameters had the robot not learned $t_i$. Doing this for each task allows us to sort the tasks by their impact on the overall performance. If, for instance, the robot uses rehearsal-methods for continuously updating its model, it then can ignore the task that leads to least harm if the task-related knowledge was dropped from the model-parameters.

## Conclusion & Further Work

This vision paper looked at the benefits of using the TEE-framework to increase the robustness and flexibility of CL-systems. We propose a starting point that can be used to answer (counter-)factual questions about CL-systems to guide

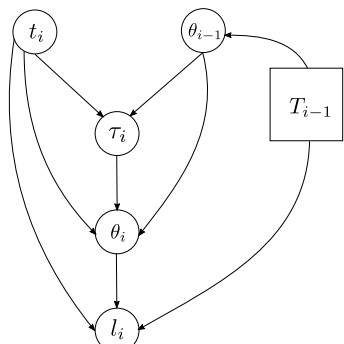

Figure 1: Continual Learning represented as a causal graph. The decision $\tau_{i-1}$ if the current model parameters $\theta_{i-1}$ are updated using task $t_i$ depend on $\theta_{i-1}$ (which is obtained) and $t_i$ only. The model-parameters $\theta_i$ at timestep $i$ influence the overall model-performance $l_i$ across all $i$ tasks.

the optimization behavior. Answering such questions is crucial in productive systems in order to give guarantees w.r.t. model-performance and to minimize computational costs (e.g. by using parameter-estimations as a warm-start). Additionally, viewing at CL-systems from a causal lens allows us to make models more transparent, e.g. by identifying knowledge that has low positive effect on the model-performance. Further work should start with solving the factual case, followed by the counterfactual case. We already have sketched a possible approach using Influence Functions which estimate the effect of samples from a dataset on the model parameters. An extension to estimate the effect of entire datasets/tasks could be employed to answer factual and counterfactual queries as shown above. Another approach could be to employ representation learning techniques which have been shown to compactly represent complex high-dimensional data in relatively low-dimensional spaces. This effectiveness could be exploited to encode parameters and tasks in compact representations. Then, the NCM framework proposed by Xia et al. (2021) could be employed to estimate CATE to answer the questions mentioned above. Also, instead of ATE other quantities such as ITE can be considered, e.g. to answer questions about specific tasks.

**Acknowledgments**   The authors thank the anonymous reviewers of the Bridge program for their valuable feedback. Furthermore, the authors acknowledge the support of the German Science Foundation (DFG) project "Causality, Argumentation, and Machine Learning" (CAML2, KE 1686/3-2) of the SPP 1999 "Robust Argumentation Machines" (RATIO). This work was supported by the Federal Ministry of Education and Research (BMBF; project "PlexPlain", FKZ 01IS19081). It benefited from the Hessian research priority programme LOEWE within the project WhiteBox, the HMWK cluster project "The Third Wave of AI" (3AI) & the National High-Performance Computing project for Computational Engineering Sciences (NHR4CES).

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
