# OpenReview forum: "Treatment Effect Estimation to Guide Model Optimization in Continual Learning"
_AAAI.org/2023/Bridge/CCBridge — AAAI23 Bridge Continual Causality_

### Official Review · Reviewer_voJN · 2022-12-01

**Rating:** 3
**Confidence:** 3

**Review:**

The paper aims to address the problem of model capacity, as well as what performance drop can be expected if we train on a new task. These questions are interesting to the community, especially the question of how to decide what neurons (weights) to unfreeze (reduce regularization ) when capacity is limited has not received much attention.

The authors propose to use techniques from "treatment effect" which refers to the causal effect of a given treatment on an outcome. However, this technique being new for me, I found the motivation and explanation of this technique insufficient to be able to say if this would contribute in a meaningful way to addressing the above questions. The authors should have spent more of the space on motivating and explanation of the reasons why these techniques could (and are the correct ones) to address these questions. I think much of the first column explanation of CL could have been skipped to reserve room for this.

Writing of article should be improved.

Minor remark: many statements could use references.

---

### Official Review · Reviewer_ZVNX · 2022-12-03
**Interesting vision-paper proposing the use of Treatment Effect Estimation for estimating when to add more resources to a model as well as for enabling graceful forgetting.**

**Rating:** 7
**Confidence:** 3

**Review:**

The paper outlines an interesting approach to combine the benefits of causal inference, particularly, with respect to Treatment Effect Estimation (TEE) for enabling efficient Continual Learning (CL) of information The paper poses two important questions for CL models to consider:

(a) When to extend neural resources/parameters to enable efficient learning of new information without degrading model performance on previous tasks?

(b) Assuming max network/resource capacity is reached, which portions of existing knowledge can be `gracefully forgotten' to make way for new learning, with the least impact on overall model performance?

I think these are important questions for CL and the authors outline an interesting and potentially very useful way to address these. I have a few questions which may be worth considering when implementing the proposed solutions.

1. For TEE in Factual Settings: One of the ways proposed to estimate the expectation of model performance ($l_i$) could be the use of Monte Carlo approaches. In practice, this can be very resource extensive to achieve. Also, to my understanding the CATE estimation may help us consider whether learning a new task with the current set of parameters might depreciate model performance or not. Perhaps, it is also important to estimate whether extending model parameters (and by how much?) benefits model performance or not. I am appreciative of the proposition but just unsure about how to practically achieve this on-the-go, sometimes on very resource-constrained devices.

2. For TEE in Counterfactual Settings: I think the proposed solutions to evaluate which task can be forgotten with the least impact on overall model performance is quite interesting. Would we need to maintain a separate matrix to record and update our estimation of ATEs after each task? How do we account for Forward Transfer, where learning one task may actually benefit subsequent tasks? And thus, impact of forgetting a task actually can be multifold?

Some Minor comments:

1. Para 1 of Motivation: "In CL the notion of a task is typically." Sentence seems incomplete.
2. Para 2 of Treatment Effect Estimation "behaved if and intervention" -> "behaved if an intervention",
3. Equation number missing for TEE in Counterfactual Settings.
4. Please include additional citations for relevant aspects of the motivation. For instance, Catastrophic Forgetting, Replay-based CL, Parameter Isolation Methods etc.

---

### Official Review · Reviewer_A4mh · 2022-12-04
**Relevant paper and Solid Framework**

**Rating:** 8
**Confidence:** 3

**Review:**

My reviews are based on these instructions sent in the email:
> Their [reviews] primary purpose is to check for factual correctness and broad relation to the bridge topics. As a guideline, we envision the reviews to be inclusive and include suggestions on the laid-out directions, rather than voicing direct critique at such an early stage.

**Relevance for the program: (High)**

The short paper is relevant to the program since it connects treatment effect estimation to model capacity in continual learning. Should the capacity be increased for new tasks? What old knowledge can be discarded while minimizing the damage? Both of these are important questions for CL and the short paper uses a TEE framework to help explore these questions.

**Factual Correctness: (Solid)**

The causal graph for the CL paradigm and the descriptions for factual/counterfactual settings seem good.

**Additional Feedback**

Even though the short paper does not include replay, I hope it inspires discussions on a causal framework encompassing replay. Despite the need for additional storage and/or generative models, there are many works showing this to be a promising direction. At this point in time, instead of discarding replay, I think it is wiser to build a more comprehensive causal framework that includes replay.

---

### Official Review · Reviewer_sEbS · 2022-12-06
**Do Calculus for ATE towards Causal Inference in CL**

**Rating:** 7
**Confidence:** 4

**Review:**

Authors propose to address the continual learning issues (i.e., updating the current parameter without losing performance on the previous task and learning the model under the fixed resources) from the perspective of Causal Inference. These issues are well-known in continual learning under the name of catastrophic forgetting and intransigence. They propose to implement causal inference using the concept of Average Treatment Effect Estimation through "Do Calculus" under the following 3 rules:
Rule 1: Decide if we can ignore an observation
Rule 2: Decide if we can treat an intervention as an observation
Rule 3: Decide if we can ignore an intervention
Another concept used by the author to measure the impact of the intervention of new tasks or samples in continual learning is the average treatment effect (ATE). The ATE compares treatments (or interventions) in randomized experiments. The author suggested that we develop a robust and flexible continual learning model based on these rules and the concept of ATE.

These concepts have been already explored and discussed various recent works, which are based on continual learning and causal inference as follows:
Hu, Xinting, et al. "Distilling causal effect of data in class-incremental learning." Proceedings of the IEEE/CVF Conference on Computer Vision and Pattern Recognition. 2021.
Javed, Khurram, Martha White, and Yoshua Bengio. "Learning causal models online." arXiv preprint arXiv:2006.07461 (2020).
Chu, Zhixuan, Stephen Rathbun, and Sheng Li. "Continual Lifelong Causal Effect Inference with Real World Evidence." (2020).

Also, the Do calculus is a well-known term for the representation of causal inference; however, it needs to be clarified how it will be additionally helpful (in the context of the works mentioned above) in continual learning.

---

### Decision · Program_Chairs · 2022-12-05

**Decision:**

Accept

**Comment:**

Accept - Poster

The paper presents a vision of guiding model optimization in continual learning, in terms of model capacity, through treatment effect estimation. The topic is highly relevant to the bridge and the reviewers generally agree that it holds prospects. Some of the constructive suggestions revolved around the need for further improvement with respect to presentation of the motivation and explanation behind the techniques being a good choice. Several references (some explicit, some in the form of implicit pointers to e.g. rehearsal) were also suggested. We encourage the authors to make use of the additional page for the camera ready to accommodate these suggestions.